# Psychological distress, fear and coping among Malaysians during the COVID-19 pandemic

**Ahmed Suparno Bahar Moni**[1]*, **Shalimar Abdullah**[2], **Mohammad Farris Iman Leong Bin Abdullah**[1], **Mohammed Shahjahan Kabir**[3], **Sheikh M. Alif**[4], **Farhana Sultana**[5,6], **Masudus Salehin**[7], **Sheikh Mohammed Shariful Islam**[8], **Wendy Cross**[7], **Muhammad Aziz Rahman**[7,9,10,11]*

1 Advanced Medical and Dental Institute, Universiti Sains Malaysia, Bertam, Kepala Batas, Penang, Malaysia, 2 Faculty of Medicine, Pusat Perubatan Universiti Kebangsaan Malaysia, Kuala Lumpur, Malaysia, 3 Faculty of Medicine, Quest International University, Ipoh, Perak, Malaysia, 4 School of Public Health and Preventive Medicine, Monash University, Melbourne, Victoria, Australia, 5 Telstra Health, Melbourne, Victoria, Australia, 6 Melbourne School of Population and Global Health, University of Melbourne, Carlton, Victoria, Australia, 7 School of Health, Federation University Australia, Berwick, Victoria, Australia, 8 Institute for Physical Activity and Nutrition, Deakin University, Burwood, Victoria, Australia, 9 Australian Institute for Primary Care and Ageing, La Trobe University, Melbourne, Victoria, Australia, 10 Department of Noncommunicable Diseases, Bangladesh University of Health Sciences (BUHS), Dhaka, Bangladesh, 11 Faculty of Public Health, Universitas Airlangga, Surabaya, Indonesia

* suparno1978@gmail.com (ASBM); ma.rahman@federation.edu.au (MAR)

## Abstract

### Introduction

The COVID-19 pandemic has enormously affected the psychological well-being, social and working life of millions of people across the world. This study aimed to investigate the psychological distress, fear and coping strategies as a result of the COVID-19 pandemic and its associated factors among Malaysian residents.

### Methods

Participants were invited to an online cross-sectional survey from Aug-Sep 2020. The study assessed psychological distress using the Kessler Psychological Distress Scale, level of fear using the Fear of COVID-19 Scale, and coping strategies using the Brief Resilient Coping Scale. Univariate and multivariate logistic regression analyses were conducted to adjust for potential confounders.

### Results

The mean age (±SD) of the participants (N = 720) was 31.7 (±11.5) years, and most of them were females (67.1%). Half of the participants had an income source, while 216 (30%) identified themselves as frontline health or essential service workers. People whose financial situation was impacted due to COVID-19 (AOR 2.16, 95% CIs 1.54–3.03), people who drank alcohol in the last four weeks (3.43, 1.45–8.10), people who were a patient (2.02, 1.39–2.93), and had higher levels of fear of COVID-19 (2.55, 1.70–3.80) were more likely to have higher levels of psychological distress. Participants who self-isolated due to exposure to COVID-19 (3.12, 1.04–9.32) and who had moderate to very high levels of psychological

**Data Availability Statement:** All relevant data are within the paper.

**Funding:** Telstra Health provided support in the form of salary for author FS. However, the authors

did not receive direct or specific funding for this work. The specific roles of this author is articulated in the 'author contributions' section.

**Competing interests:** The authors have read the journal's policy and have the following competing interests: RS is a staff member of Telstra Health (https://www.telstrahealth.com/). There are no patents, products in development or marketed products associated with this research to declare. This does not alter our adherence to PLOS ONE policies on sharing data and materials.

distress (2.56, 1.71–3.83) had higher levels of fear. Participants who provided care to a family member/patient with a suspected case of COVID-19 were more likely to be moderately to highly resilient compared to those who did not.

## Conclusion

Vulnerable groups of individuals such as patients and those impacted financially during COVID-19 should be supported for their mental wellbeing. Behavioural interventions should be targeted to reduce the impact of alcohol drinking during such crisis period.

## Introduction

The world is currently facing a pandemic due to the rapid spread of coronavirus disease 2019 (COVID-19), caused by severe acute respiratory syndrome coronavirus 2 (SARS-CoV-2). As of 23rd March 2021, an estimated 124,291,475 confirmed cases and around 2,735,205 deaths have been attributed to COVID-19 affecting more than 219 countries and territories across the world [1]. Malaysia has reported 334,156 confirmed cases with 1,238 cumulative deaths with a case fatality rate of 0.4% [1]. Although the case fatality rate was low in Malaysia compared to other developed countries like USA or UK, people were anxious as the virus could spread rapidly from one person to another through direct or indirect contact [2].

In Malaysia, the first COVID-19 case was detected on 25 January 2020 [3]. With a surge in cases thereafter, physical distancing rules, restrictions on social gatherings, appropriate use of face masks, Movement Control Order (MCO), Conditional Movement Control Order (CMCO), extended movement control order, and border closures were implemented by the Malaysian Government between mid-March to August-2020 to curb the spread of the disease [4]. However, Malaysia has seen a resurgence of COVID-19 cases and is currently facing a third wave of infection and is under the second CMCO from 9-November-2020 in all states except Perlis, Kelantan, Pahang and Sarawak. Malaysia has launched COVID-19 vaccination program on the 24th February 2021 [5]. The impact of all these spatial distancing policies and the uncertainty of returning to normalcy have direct and indirect impact on social life as well as mental wellbeing of the community people. Those interim actions such as MCOs or lockdowns, physical distancing and quarantine have reportedly led to heightened fears, stress and anxiety amongst individuals globally.

A recent review found women, younger individuals, those living in rural areas, those with lower socioeconomic status, those at higher-risk of COVID-19 infection and longer media exposure to be associated with higher levels of anxiety and depression [6]. Individual studies have shown that the COVID-19 pandemic affected people in different countries in different ways with some groups being more vulnerable than others. In Australia, pre-existing mental health conditions, increased smoking and alcohol during the lockdown and high levels of fear and being female were associated with higher levels of psychological distress [7]. Similarly, in the UK, females, younger age, lower annual income, smokers and co-morbidity were associated with poor mental health [8]. While in Italy, female gender, negative affect and detachment were associated with higher levels of stress [9]. In some studies in China, frequent and prolonged social media exposure during the COVID-19 pandemic was found to be strongly associated with anxiety and depression [10].

Previous studies have reported the negative influence of pandemics on psychological wellbeing, which can lead to acute depression and anxiety [7, 11]. Evidence suggests that frontline

healthcare workers, who were directly involved in the collection of samples, diagnosis, treatment, and care of patients during an outbreak were at higher risk of developing psychological distress and mental health symptoms [12]. Previous evidence documented immediate psychological impacts amongst frontline healthcare workers with symptoms of anxiety, distress, depression, fear of spreading infection to family, friends and colleagues [7, 13]. Lower sleep quality due to anxiety and stress, which eventually reduced self-efficacy exponentially among the medical staff has also been reported [14].

Only recently studies have emerged to show the negative impact of the pandemic on children, older people, pregnant women, university students, people with weight issues and the general population as a whole. An Iranian study revealed effect of fear of COVID-19 was significantly associated with depression, anxiety, suicidal intention and mental quality of life among the pregnant women [15]. In a study with older people, fear of COVID-19 significantly mediated the association between perceived health status, and insomnia, mental health and COVID-19 preventative behaviours [16]. A recent study among university students from Indonesia, Taiwan and Thailand found that Thai students had the highest level of anxiety but limited resources to fight the COVID-19 pandemic, whereas Taiwanese students were more negatively affected by information gathering from the internet; such less perceived satisfactory support was associated with more suicidal thoughts among Indonesian students [17]. Stressors of COIVD-19 pandemic could also result in behaviour impairments of children and adolescents, which could potentially impact psychological wellbeing in early life and adulthood [18].

Studies on the impact of COVID-19 on mental health are limited in Malaysia and most of them were conducted amongst students. In one study using online survey, out of 983 Malaysian students, 20.4%, 6.6%, and 2.8% experienced minimal to moderate, marked to severe, and most extreme levels of anxiety, respectively. Female gender, age under 18 years, pre-university level of education, management studies, and staying alone were significantly associated with higher levels of anxiety. The main stressors included financial constraints, remote online teaching, and uncertainty about the future regarding study and career also affecting the mental health [2]. In another study amongst Malaysian university students, the prevalence of anxiety was much higher; 30.5% were experiencing mild, 31.1% moderate, and 26.1% severe anxiety; age >20 years, Chinese ethnicity, decreased family income, co-morbid conditions, and spending time watching COVID-related news and infected friends and relatives were found to be associated with increased anxiety [19]. In another study in Malaysia age <25 years and females were more likely to have higher levels of fear of COVID-19; however, 70% of the respondents were also students in this study [20].

There is limited evidence regarding the impact of COVID-19 on psychological distress, fear and coping strategies as a whole and amongst community members and healthcare workers in Malaysia. We, therefore, conducted this study to understand the extent of the mental health burden in the community settings in Malaysia during the COVID-19 pandemic. The study will identify population subgroups more at risk of developing poor mental health outcomes and enable policy makers to guide resource planning and design psychosocial interventions targeted to these high-risk and vulnerable groups of population.

## Materials and methods

### Study design and settings

A cross-sectional study was conducted between August and September 2020. An online survey link was shared in different online platforms, including Facebook, Twitter and LinkedIn inviting online users to participate in this study.

## Study population

Study participants included patients, university students and healthcare professionals residing in Malaysia. To be eligible, participants had to be 18 years or above and were literate enough to respond to an online questionnaire in English. The participants who took <1 minute to complete the questionnaire, were excluded during analyses.

## Sampling

Sample size was calculated using OpenEpi [21]. Considering 32.6 million population of Malaysia [22], 30% estimated prevalence of stress amongst Malaysians [23, 24], at 95% confidence intervals and 80% power, the estimated minimum sample size was 323. Snowball sampling technique was used to recruit the study participants. Once any participant filled up the online questionnaire, h/she forwarded the survey link to own personal/professional networks.

## Data collection

Google form was used to develop the study questionnaire. The first page included participant information statement and the consent form. Participants, who provided consents, could move to the next screen. There were two screening questions to determine eligibility of the study participants, one was age and the other was location of residence. Eligible participants accessed the full study questionnaire and responses were collected anonymously. The online survey link was shared through university/hospital staff/students' emails, text messages, WhatsApp and other social media platforms such as Facebook, Twitter and LinkedIn. Patients visiting any healthcare settings or university students within the defined study period were informed about the study and of the online link by the respective healthcare professionals or university faculty members.

## Study tool

We used the same survey questionnaire (except residence location/region in Malaysia) which was used earlier by the Australian investigators included in this study [7]. Three validated tools were included in the survey questionnaire. The Kessler Psychological Distress Scale (K10) tool having ten items was used to assess psychological distress [25], the Fear of COVID-19 scale (FCV-19S) having seven items was used to assess the levels of fear [26], and the Brief Resilient Coping Scale (BRCS) having four items was used to assess the levels of coping [27]. Each of those tools collected responses using a 5-point likert scale and the scoring was categorised as discussed in earlier study [7]. Reliability of using these tools had also been examined in a recent study [28]. The questionnaire was pre-tested and no changes were made.

## Data analyses

Data from Google forms were downloaded and analysed using STATA v.12. Continuous variables were described using descriptive statistics such as mean standard deviations, and proportions. Scoring in the K10 scale was re-defined into low (score 10–15) and moderate to very high (score 16–50), the FCV-19S scale to low (score 7–21) and high (score 22–35) and BRCS scale categorised into low (score 4–13) and medium to high (score 14–20) resilient coping. We used univariate and multivariate logistic regression to investigate the associations. The multivariate models were adjusted for socio-demographic variables such as age, gender, living status, country of birth, education, and employment status.

### Ethics

Ethics approval was obtained from the Human Research Ethics Committee (HREC) at Universiti Sains Malaysia (USM/JEPeM/COVID19-40). Data were collected anonymously and could not be linked back to identify any participant. Contact details of Befrienders was included at the end of the online questionnaire, allowing participant/s to access necessary support in case of distress during filling questionnaire.

## Results

A total of 720 individuals participated in this study. Mean age (±SD) of the participants was 31.7 (±11.5) years, and most of them (56.7%) were in the age group 18–29 years. More than two-thirds of the participants (67.1%) were females. A quarter of the study population (27.1%) was from Penang, and another quarter (22.9%) was from Perak in Malaysia. Almost all of them (90.8%) were born in Malaysia. A third of the study population (30%) identified themselves as frontline or essential service workers, and a third (31.9%) was identified as patients. Details of the characteristics of the study population are presented in Table 1.

About two-thirds of the study participants (62.1%) experienced moderate to very high levels of psychological distress. Only a quarter (27.1%) reported high levels of fear of COVID-19 and two-thirds of the participants (65.1%) were identified as having medium to high resilient coping (Tables 2–4).

Table 5 shows the univariate and multivariate analyses regarding factors associated with psychological distress. Moderate to very high levels of psychological distress was associated with impacted financial situation due to COVID-19 (AOR 2.16, 95% CIs 1.54–3.03, p<0.001), alcohol drinking in the last four weeks (AOR 3.43, 95% CIs 1.45–8.10, p<0.01), being a patient (AOR 2.02, 95% CIs 1.39–2.93, p<0.001), and having higher levels of fear of COVID-19 (AOR 2.55, 95% CIs 1.70–3.80, p<0.001). However, those in the older age groups i.e. 30–59 years (AOR 0.51, 95% CIs 0.27–0.95, p<0.05), those of ≥60 years old (AOR 0.07, 95% CIs 0.01–0.37, p<0.01) and those who had medium to highly resilient coping (AOR 0.54, 95% CIs 0.38–0.77, p<0.01) were less likely experience higher psychological distress (Table 5).

Table 6 shows the univariate and multivariate analyses regarding factors associated with fear of COVID-19. Study participants who had been tested negative for COVID-19 but were self-isolating (AOR 3.12, 95% CIs 1.04–9.32, p<0.05) and those who had moderate to very high levels of psychological distress (AOR 2.56, 95% CIs 1.71–3.83, p<0.001) also had high levels of fear. Conversely, study participants who were born in Malaysia (AOR 0.39, 95% CIs 0.18–0.86, p<0.05) and who drank alcohol in the last four weeks (AOR 0.26, 95% CIs 0.10–0.68, p<0.01) had lower levels of fear in this study (Table 6).

Study participants who provided care to a family member/patient with known/suspected case of COVID-19 had medium to high resilient coping (AOR 1.87, 95% CIs 1.01–3.46, p<0.05), whereas participants with moderate to very high levels of psychological distress had low resilient coping (AOR 0.54, 95% CIs 0.38–0.76, p<0.01) (Table 7).

## Discussion

This cross-sectional survey found that a large proportion of Malaysian residents experienced moderate to very high levels of psychological distress as a result of the COVID-19 pandemic. Malaysians, whose financial situation was impacted by COVID-19, those who drank alcohol in the past four weeks, those who self-identified as patients and those with higher levels of fear, were more likely to experience higher psychological distress. Higher levels of psychological distress were also associated with higher levels of fear and so were people who self-identified as patients. A large majority of the participants also reported as having medium to highly resilient

**Table 1. Characteristics of the study population.**

| Characteristics | Total, n(%) |
| --- | --- |
| Total study participants | 720 |
| **Age (in years)** | **702** |
| Mean (±SD) | 31.7 (11.5) |
| Range | 19 to 76 |
| **Age groups** | **702** |
| 18–29 years | 398 (56.7) |
| 30–59 years | 282 (40.2) |
| ≥60 years | 22 (3.1) |
| **Gender** | **720** |
| Male | 235 (32.6) |
| Female | 483 (67.1) |
| Others | 2 (0.3) |
| **Location in Malaysia** | **720** |
| Johor | 24 (3.3) |
| Kedah | 50 (6.9) |
| Kelantan | 60 (8.3) |
| Kuala Lumpur | 72 (10.0) |
| Kuala Terengganu | 5 (0.7) |
| Malacca | 7 (1.0) |
| Nigari Sembilan | 8 (1.1) |
| Pahang | 16 (2.2) |
| Penang | 195 (27.1) |
| Perak | 165 (22.9) |
| Perlis | 3 (0.4) |
| Sabah | 13 (1.8) |
| Sarawak | 6 (0.8) |
| Selangor | 96 (13.3) |
| **Living status** | **718** |
| Live without family members (on your own/shared house/others) | 136 (18.9) |
| Live with family members (partner and/or children) | 559 (77.9) |
| **Born in Malaysia** | **720** |
| No | 66 (9.2) |
| Yes | 654 (90.8) |
| **Completed level of education** | **716** |
| Secondary | 113 (15.8) |
| Diploma | 119 (16.6) |
| Degree (Bachelor) | 301 (42.0) |
| Masters and above | 183 (25.6) |
| **Current employment condition** | **710** |
| Unemployed/Home duties (No source of income) | 309 (43.5) |
| Jobs affected by COVID-19 (lost job/working hours reduced/afraid of job loss) | 46 (6.5) |
| Have an income source (employed/Government benefits) | 355 (50.0) |
| **Perceived distress due to change of employment status** | **699** |
| A little to none | 482 (69.0) |
| Moderate to a great deal | 217 (31.0) |
| **Self-identification as a frontline or essential service worker** | **720** |
| No | 504 (70.0) |

(*Continued*)

**Table 1.** (Continued)

| Characteristics | Total, n(%) |
|---|---|
| Yes | 216 (30.0) |
| **COVID-19 impacted financial situation** | **720** |
| No | 379 (52.6) |
| Yes | 341 (47.4) |
| **Co-morbidities** | **612** |
| No | 478 (78.1) |
| Psychiatric/Mental health problem | 20 (3.3) |
| Other co-morbidities* | 114 (18.6) |
| **Smoking** | **720** |
| Never smoker | 656 (91.1) |
| Ever smoker (Daily/Non-daily/Ex) | 64 (8.9) |
| **Current alcohol drinking (last 4 weeks)** | **713** |
| No | 665 (93.3) |
| Yes | 48 (6.7) |
| **Increased alcohol drinking over the last 4 weeks** | **48** |
| No | 34 (70.8) |
| Yes | 14 (29.2) |
| **Provided care to a family member/patient with known/suspected case of COVID-19** | **715** |
| No | 647 (90.5) |
| Yes | 68 (9.5) |
| **Experience related to COVID-19 pandemic (multiple responses possible)** | **688** |
| No known exposure to COVID-19 | 638 (92.7) |
| I had recent overseas travel history and was in self-quarantine | 10 (1.5) |
| I have been tested negative for COVID-19 but self-isolating | 40 (5.8) |
| **Self-identification as a patient (visited a healthcare provider in the last 4 weeks)** | **715** |
| No | 487 (68.1) |
| Yes | 228 (31.9) |
| **Healthcare service use in the last 4 weeks** | **301** |
| Telehealth consultation/Use of national helpline | 240 (79.7) |
| In-person visit to a healthcare provider | 45 (15.0) |
| Used both services | 16 (5.3) |
| **Healthcare service use to overcome COVID-19 related stress in the last 4 weeks** | **707** |
| No | 689 (97.5) |
| Yes | 18 (2.5) |

* Stroke/Hypertension/Hyperlipidaemia/Diabetes/Cancer/Chronic respiratory illness

coping during this pandemic especially those who provided care to family members affected by the pandemic.

Findings of our survey in Malaysia are comparable to similar studies conducted in other parts of the globe. Financial difficulty is associated with anxiety as well as a predisposition to depression after several months of quarantine exacerbated by undue uncertainty [29]. Studies among the general population in China and India have shown that poor economic status and difficulties in meeting living expenses during the COVID-19 pandemic significantly increasing the degree of psychological distress [30, 31]. Likewise, studies during the SARS and MERS epidemics have also shown that increased psychological distress was associated with increased financial difficulties. This could be explained by the emergence of a sense of uncertainty and

**Table 2. Level of psychological distress among the study participants.**

| Anxiety and Depression Checklist (K10) (last 4 weeks) | Total, n(%) |
|---|---|
| **About how often did you feel tired out for no good reason?** | **720** |
| None | 184 (25.6) |
| A little | 182 (25.3) |
| Sometime | 248 (34.4) |
| Most of the time | 83 (11.5) |
| All the time | 23 (3.2) |
| **About how often did you feel nervous?** | **720** |
| None | 229 (31.8) |
| A little | 221 (30.7) |
| Sometime | 206 (28.6) |
| Most of the time | 55 (7.6) |
| All the time | 9 (1.3) |
| **About how often did you feel so nervous that nothing could calm you down?** | **720** |
| None | 378 (52.5) |
| A little | 178 (24.7) |
| Sometime | 129 (17.9) |
| Most of the time | 31 (4.3) |
| All the time | 4 (0.6) |
| **About how often did you feel hopeless?** | **720** |
| None | 361 (50.1) |
| A little | 176 (24.4) |
| Sometime | 120 (16.7) |
| Most of the time | 51 (7.1) |
| All the time | 12 (1.7) |
| **About how often did you feel restless or fidgety?** | **720** |
| None | 296 (41.1) |
| A little | 200 (27.8) |
| Sometime | 164 (22.8) |
| Most of the time | 46 (6.4) |
| All the time | 14 (1.9) |
| **About how often did you feel so restless you could not sit still?** | **720** |
| None | 382 (53.1) |
| A little | 175 (24.3) |
| Sometime | 134 (18.6) |
| Most of the time | 22 (3.1) |
| All the time | 7 (1.0) |
| **About how often did you feel so depressed?** | **720** |
| None | 310 (43.1) |
| A little | 205 (28.5) |
| Sometime | 139 (19.3) |
| Most of the time | 48 (6.7) |
| All the time | 18 (2.5) |
| **About how often did you feel that everything was an effort?** | **720** |
| None | 194 (26.9) |
| A little | 224 (31.1) |
| Sometime | 178 (24.7) |
| Most of the time | 98 (13.6) |

(*Continued*)

**Table 2.** (Continued)

| Anxiety and Depression Checklist (K10) (last 4 weeks) | Total, n(%) |
|---|---|
| All the time | 26 (3.6) |
| **About how often did you feel so sad that nothing could cheer you up?** | **720** |
| None | 325 (45.1) |
| A little | 194 (26.9) |
| Sometime | 144 (20.0) |
| Most of the time | 42 (5.8) |
| All the time | 15 (2.1) |
| **About how often did you feel worthless?** | **720** |
| None | 378 (52.5) |
| A little | 178 (24.7) |
| Sometime | 107 (14.9) |
| Most of the time | 34 (4.7) |
| All the time | 23 (3.2) |
| **K10 score (total)** | **720** |
| Mean (±SD) | 20.0 (8.3) |
| Range | 10 to 50 |
| **Level of psychological distress (K10 categories)** | **720** |
| Low (score 10–15) | 273 (37.9) |
| Moderate (score 16–21) | 177 (24.6) |
| High (score 22–29) | 151 (21.0) |
| Very high (score 30–50) | 119 (16.5) |

lack of security during the pandemic [32]. Hence, our finding further supports the inverse association between increased financial difficulties during COVID-19 and the occurrence of psychological distress.

In line with our findings, Ahmed et al. conducted a study in a Chinese population where they had also reported high prevalence of alcohol use and alcohol dependence during the COVID-19 pandemic [33]. Given that this was a cross-sectional study, it was possible that psychological distress led to increased alcohol use as a coping mechanism to deal with COVID-19 induced psychological distress, but the converse was also likely that increased alcohol use worsened psychological distress [34].

This study also showed that people who self-identified as a patient i.e. having visited a healthcare provider in the past four weeks, were more likely to experience higher psychological distress. However, it was not clear from the survey questionnaire if patients had visited a healthcare provider for COVID-19 like symptoms or for other medical conditions. Being infected with COVID-19 or awaiting the possibility of becoming ill was likely to be more stressful because of the fear of mortality or morbidity associated with a disease [29]. Those infected with COVID-19 had higher levels of depression, anxiety, and post-traumatic stress symptoms when compared to those not infected. In fact, people with a history of being infected with COVID-19 had reported unresolved fear, guilt, and helplessness. They were likely to be affected by the stigma of being labelled as someone who had been infected and faced uncertainty about their prognosis and future [35]. Moreover, the findings of this study also highlighted that those who tested negative for COVID-19 but maintained self-isolation from others had higher levels of fear, and those with higher levels of fear of COVID-19 also had moderate to high psychological distress. Knowing the high infectivity capability of the virus, the asymptomatic presentation of some of the COVID-19 positive cases, and the consequences

**Table 3. Level of fear of COVID-19 among the study participants.**

| Fear of COVID-19 Scale (FCV-19S) individual items | Total, n(%) |
|---|---|
| **I am most afraid of COVID-19** | **720** |
| Strongly disagree | 106 (14.7) |
| Somewhat disagree | 109 (15.1) |
| Neither agree nor disagree | 150 (20.8) |
| Somewhat agree | 237 (32.9) |
| Strongly agree | 118 (16.4) |
| **It makes me uncomfortable to think about COVID-19** | **720** |
| Strongly disagree | 136 (18.9) |
| Somewhat disagree | 118 (16.4) |
| Neither agree nor disagree | 165 (22.9) |
| Somewhat agree | 230 (31.9) |
| Strongly agree | 71 (9.9) |
| **My hands become clammy when I think about COVID-19** | **720** |
| Strongly disagree | 333 (46.3) |
| Somewhat disagree | 166 (23.1) |
| Neither agree nor disagree | 164 (22.8) |
| Somewhat agree | 42 (5.8) |
| Strongly agree | 15 (2.1) |
| **I am afraid of losing my life because of COVID-19** | **720** |
| Strongly disagree | 188 (26.1) |
| Somewhat disagree | 113 (15.7) |
| Neither agree nor disagree | 154 (21.4) |
| Somewhat agree | 180 (25.0) |
| Strongly agree | 85 (11.8) |
| **When watching news and stories about COVID-19 on social media, I become nervous or anxious** | **720** |
| Strongly disagree | 165 (22.9) |
| Somewhat disagree | 139 (19.3) |
| Neither agree nor disagree | 161 (22.4) |
| Somewhat agree | 216 (30.0) |
| Strongly agree | 39 (5.4) |
| **I cannot sleep because I'm worrying about getting COVID-19** | **720** |
| Strongly disagree | 413 (57.4) |
| Somewhat disagree | 135 (18.8) |
| Neither agree nor disagree | 129 (17.9) |
| Somewhat agree | 33 (4.6) |
| Strongly agree | 10 (1.4) |
| **My heart races or palpitates when I think about getting COVID-19** | **720** |
| Strongly disagree | 346 (48.1) |
| Somewhat disagree | 123 (17.1) |
| Neither agree nor disagree | 142 (19.7) |
| Somewhat agree | 90 (12.5) |
| Strongly agree | 19 (2.6) |
| **FCV-19S score (total)** | **720** |
| Mean (±SD) | 17.5 (6.3) |
| Range | 7 to 35 |
| **Level of fear of COVID-19 (FCV-19S categories)** | **720** |
| Low (score 7–21) | 525 (72.9) |
| High (score 22–35) | 195 (27.1) |

**Table 4. Coping during COVID-19 pandemic among the study participants.**

| Brief Resilient Coping Scale (BRCS) individual items | Total, n(%) |
|---|---|
| **I look for creative ways to alter difficult situations** | **720** |
| Does not describe me at all | 24 (3.3) |
| Does not describe me | 55 (7.6) |
| Neutral | 289 (40.1) |
| Describes me | 270 (37.5) |
| Describes me very well | 82 (11.4) |
| **Regardless of what happens to me, I believe I can control my reaction to it** | **720** |
| Does not describe me at all | 7 (1.0) |
| Does not describe me | 56 (7.8) |
| Neutral | 249 (34.6) |
| Describes me | 302 (41.9) |
| Describes me very well | 106 (14.7) |
| **I believe I can grow in positive ways by dealing with difficult situations** | **720** |
| Does not describe me at all | 10 (1.4) |
| Does not describe me | 33 (4.6) |
| Neutral | 182 (25.3) |
| Describes me | 348 (48.3) |
| Describes me very well | 147 (20.4) |
| **I actively look for ways to replace the losses I encounter in life** | **720** |
| Does not describe me at all | 26 (3.6) |
| Does not describe me | 40 (5.6) |
| Neutral | 281 (39.0) |
| Describes me | 304 (42.2) |
| Describes me very well | 69 (9.6) |
| **BRCS score (total)** | **720** |
| Mean (±SD) | 14.4 (2.7) |
| Range | 4 to 20 |
| **Level of coping (BRCS categories)** | **720** |
| Low resilient coping (score 4–13) | 251 (34.9) |
| Medium resilient coping (score 14–16) | 345 (47.9) |
| High resilient coping (score 17–20) | 124 (17.2) |

of the COVID-19 infection had created enormous fear among the general population and healthcare workers [36–38]. Unresolved fear which led to long-lasting stress might have predisposed individuals to psychological distress during the COVID-19 pandemic [39]. Hence, our study further strengthened the relationship between fear of the COVID-19 pandemic and increased psychological distress.

Factors identified as protective factors against psychological distress in this study was older age (≥30 years) and having higher level of resilience. Several studies on the psychological impact of COVID-19 in the general population reported that younger people (aged 21 to 40 years) were at higher risk of predisposing to depression and anxiety [40] highlighting consistency with other studies. Younger people may have greater focus on COVID-19 and higher degree of worry about the spread of COVID-19 presumably because of more and/or frequent access to news/social media, hence increasing their risk of psychological distress compared to older people [40].

Those with higher resilience, particularly in the components of tenacity, strength, and optimism, have shown to experience less mental health complications during the COVID-19

**Table 5. Factors associated with high psychological distress among the study population (based on K10 scoring).**

| Characteristics | Moderate to Very High (score 16–50), n(%) | Low (score 10–15), n(%) | Unadjusted analyses | | | Adjusted analyses | | |
|---|---|---|---|---|---|---|---|---|
| | | | p | OR | 95% CIs | p | AOR | 95% CIs |
| Total study participants | 447 | 273 | | | | | | |
| **Age groups** | **433** | **269** | | | | | | |
| 18–29 years | 278 (64.2) | 120 (44.6) | | 1 | | | 1 | |
| 30–59 years | 153 (35.3) | 129 (48.0) | *<0.001* | *0.51* | *0.37–0.70* | *0.035* | *0.51* | *0.27–0.95* |
| ≥60years | 2 (0.5) | 20 (7.4) | *<0.001* | *0.04* | *0.01–0.19* | *0.002* | *0.07* | *0.01–0.37* |
| **Gender** | **445** | **273** | | | | | | |
| Male | 126 (28.3) | 109 (39.9) | | 1 | | | 1 | |
| Female | 319 (71.17) | 164 (60.1) | *0.001* | *1.68* | *1.22–2.31* | 0.186 | 1.57 | 0.80–3.07 |
| **Living status** | **426** | **269** | | | | | | |
| Live without family members (on your own/shared house/others) | 90 (21.1) | 46 (17.1) | | 1 | | | 1 | |
| Live with family members (partner and/or children) | 336 (78.9) | 223 (82.9) | 0.193 | 0.77 | 0.52–1.14 | 0.729 | 0.80 | 0.23–2.79 |
| **Born in Malaysia** | **447** | **273** | | | | | | |
| No | 53 (11.9) | 13 (4.8) | | 1 | | | 1 | |
| Yes | 394 (88.1) | 260 (95.2) | *0.001* | *0.37* | *0.20–0.70* | 0.257 | 0.60 | 0.24–1.46 |
| **Completed level of education** | **444** | **272** | | | | | | |
| Secondary | 76 (17.1) | 37 (13.6) | | 1 | | | 1 | |
| Diploma | 76 (17.1) | 43 (15.8) | 0.587 | 0.86 | 0.50–1.48 | 0.744 | 0.78 | 0.17–3.54 |
| Degree (Bachelor) | 209 (47.1) | 92 (33.8) | 0.670 | 1.11 | 0.70–1.76 | 0.466 | 0.52 | 0.09–2.99 |
| Masters and above | 83 (18.7) | 100 (36.8) | *<0.001* | *0.40* | *0.25–0.66* | 0.466 | 0.66 | 0.22–2.01 |
| **Current employment condition** | **441** | **269** | | | | | | |
| Unemployed/Home duties | 167 (37.9) | 142 (52.8) | | 1 | | | 1 | |
| Jobs affected by COVID-19 (lost job/working hours reduced/afraid of job loss) | 29 (6.6) | 17 (6.3) | 0.254 | 1.45 | 0.77–2.75 | 0.698 | 1.24 | 0.42–3.69 |
| Have an income source (employed/Government benefits) | 245 (55.6) | 110 (40.9) | *<0.001* | *1.89* | *1.38–2.60* | 0.887 | 1.08 | 0.36–3.27 |
| **Perceived distress due to change of employment status** | **434** | **265** | | | | | | |
| A little to none | 319 (73.5) | 163 (61.5) | | 1 | | | 1 | |
| Moderate to a great deal | 115 (26.5) | 102 (38.5) | *0.001* | *0.58* | *0.42–0.80* | 0.981 | NA | NA |
| **Self-identification as a frontline or essential service worker** | **447** | **273** | | | | | | |
| No | 289 (64.7) | 215 (78.8) | | 1 | | | 1 | |
| Yes | 158 (35.3) | 58 (21.2) | *<0.001* | *2.03* | *1.43–2.87* | 0.071 | 1.59 | 0.96–2.63 |
| **COVID-19 impacted financial situation** | **447** | **273** | | | | | | |
| No | 200 (44.7) | 179 (65.6) | | 1 | | | 1 | |
| Yes | 247 (55.3) | 94 (34.4) | *<0.001* | *2.35* | *1.72–3.21* | *0.000* | *2.16* | *1.54–3.03* |
| **Co-morbidities** | **372** | **240** | | | | | | |
| No | 274 (73.7) | 204 (85.0) | | 1 | | | 1 | |

(*Continued*)

**Table 5.** (Continued)

| Characteristics | Moderate to Very High (score 16–50), n(%) | Low (score 10–15), n(%) | Unadjusted analyses | | | Adjusted analyses | | |
|---|---|---|---|---|---|---|---|---|
| | | | p | OR | 95% CIs | p | AOR | 95% CIs |
| Psychiatric/Mental health problem | 10 (2.7) | 10 (4.2) | 0.518 | 0.74 | 0.30–1.82 | 0.254 | 0.24 | 0.02–2.79 |
| Other co-morbidities* | 88 (23.7) | 26 (10.8) | <0.001 | 2.52 | 1.57–4.05 | 0.285 | 1.47 | 0.73–2.97 |
| **Smoking** | **447** | **273** | | | | | | |
| Never smoker | 405 (90.6) | 251 (91.9) | | 1 | | | 1 | |
| Ever smoker (Daily/Non-daily/Ex) | 42 (9.4) | 22 (8.1) | 0.541 | 1.18 | 0.69–2.03 | 0.734 | 0.83 | 0.28–2.47 |
| **Current alcohol drinking (last 4 weeks)** | **443** | **270** | | | | | | |
| No | 403 (91.0) | 262 (97.0) | | 1 | | | 1 | |
| Yes | 40 (9.0) | 8 (3.0) | 0.002 | 3.25 | 1.50–7.06 | 0.005 | 3.43 | 1.45–8.10 |
| **Provided care to a family member/patient with known/ suspected case of COVID-19** | **443** | **272** | | | | | | |
| No | 394 (88.9) | 253 (93.0) | | 1 | | | 1 | |
| Yes | 49 (11.1) | 19 (7.0) | 0.071 | 1.66 | 0.95–2.88 | 0.120 | 1.60 | 0.89–2.89 |
| **Experience related to COVID-19 pandemic** | **423** | **265** | | | | | | |
| No known exposure to COVID-19 | 385 (91.0) | 253 (95.5) | | 1 | | | 1 | |
| I had recent overseas travel history and was in self-quarantine | 7 (1.7) | 3 (1.1) | 0.538 | 1.53 | 0.39–5.98 | NA | NA | NA |
| I have been tested negative for COVID-19 but self-isolating | 31 7.3) | 9 (3.4) | 0.035 | 2.26 | 1.06–4.83 | 0.592 | 0.74 | 0.25–2.23 |
| **Self-identification as a patient (visited a healthcare provider in the last 4 weeks)** | **444** | **271** | | | | | | |
| No | 280 (63.1) | 207 (76.4) | | 1 | | | 1 | |
| Yes | 164 (36.9) | 64 (23.6) | <0.001 | 1.89 | 1.35–2.66 | 0.000 | 2.02 | 1.39–2.93 |
| **Healthcare service use in the last 4 weeks** | **207** | **94** | | | | | | |
| Telehealth consultation/Use of national helpline | 34 (16.4) | 11 (11.7) | | 1 | | | 1 | |
| In-person visit to a healthcare provider | 160 (77.3) | 80 (85.1) | 0.243 | 0.65 | 0.31–1.34 | 0.473 | 0.76 | 0.35–1.62 |
| Used both services | 13 (6.3) | 3 (3.2) | 0.643 | 1.40 | 0.34–5.84 | 0.551 | 1.57 | 0.36–6.84 |
| **Level of fear of COVID-19 (FCV-19S categories)** | **447** | **273** | | | | | | |
| Low (score 7–21) | 301 (67.3) | 224 (82.1) | | 1 | | | 1 | |
| High (score 22–35) | 146 (32.7) | 49 (17.9) | <0.001 | 2.22 | 1.54–3.20 | 0.000 | 2.55 | 1.70–3.80 |
| **Level of coping (BRCS categories)** | **447** | **273** | | | | | | |
| Low resilient coping (score 4–13) | 176 (39.4) | 75 (27.5) | | 1 | | | 1 | |
| Medium to high resilient coping (score 14–20) | 271 (60.6) | 198 (72.5) | 0.001 | 0.58 | 0.42–0.81 | 0.001 | 0.54 | 0.38–0.77 |
| **Healthcare service use to overcome COVID-19 related stress in the last 4 weeks** | **440** | **267** | | | | | | |
| No | 426 (96.8) | 263 (98.5) | | 1 | | | 1 | |
| Yes | 14 (3.2) | 4 (1.5) | 0.168 | 2.16 | 0.70–6.63 | 0.196 | 2.13 | 0.68–6.69 |

Adjusted for: age, gender, living status, born in Malaysia, education and employment

* Cardiac diseases/Stroke/Hypertension/Hyperlipidemia/Diabetes/Cancer/Chronic respiratory disease

**Table 6. Factors associated with high levels of fear of COVID-19 among the study population (based on FCV-19S scoring).**

| Characteristics | High (score 22–35), n(%) | Low (score 7–21), n(%) | Unadjusted analyses | | | Adjusted analyses | | |
|---|---|---|---|---|---|---|---|---|
| | | | p | OR | 95% CIs | p | AOR | 95% CIs |
| Total study participants | 195 | 525 | | | | | | |
| **Age groups** | **191** | **511** | | | | | | |
| 18–29 years | 105 (55.0) | 293 (57.3) | | 1 | | | 1 | |
| 30–59 years | 81 (42.4) | 201 (39.3) | 0.500 | 1.12 | 0.80–1.58 | 0.493 | 1.25 | 0.66–2.39 |
| ≥60 years | 5 (2.6) | 17 (3.3) | 0.705 | 0.82 | 0.30–2.28 | 0.599 | 0.70 | 0.18–2.67 |
| **Gender** | **195** | **523** | | | | | | |
| Male | 65 (33.3) | 170 (32.5) | | 1 | | | 1 | |
| Female | 130 (66.7) | 353 (67.5) | 0.833 | 0.96 | 0.68–1.37 | 0.795 | 1.10 | 0.54–2.26 |
| **Living status** | **187** | **508** | | | | | | |
| Live without family members (on your own/shared house/others) | 38 (20.3) | 98 (19.3) | | 1 | | | 1 | |
| Live with family members (partner and/or children) | 149 (79.7) | 410 (80.7) | 0.762 | 0.94 | 0.62–1.43 | 0.276 | 2.53 | 0.48–13.5 |
| **Born in Malaysia** | **195** | **525** | | | | | | |
| No | 25 (12.8) | 41 (7.8) | | 1 | | | 1 | |
| Yes | 170 (87.2) | 484 (92.2) | *0.038* | *0.58* | *0.34–0.98* | *0.020* | *0.39* | *0.18–0.86* |
| **Completed level of education** | **194** | **522** | | | | | | |
| Secondary | 35 (18.0) | 78 (14.9) | | 1 | | | 1 | |
| Diploma | 29 (14.9) | 90 (17.2) | 0.262 | 0.72 | 0.40–1.28 | 0.239 | 0.32 | 0.05–2.14 |
| Degree (Bachelor) | 81 (41.8) | 220 (42.1) | 0.413 | 0.82 | 0.51–1.32 | 0.251 | 0.29 | 0.04–2.38 |
| Masters and above | 49 (25.3) | 134 (25.7) | 0.437 | 0.81 | 0.49–1.37 | 0.152 | 0.32 | 0.07–1.52 |
| **Current employment condition** | **96** | **259** | | | | | | |
| Unemployed/Home duties | 88 (45.8) | 221 (42.7) | | 1 | | | 1 | |
| Jobs affected by COVID-19 (lost job/working hours reduced/afraid of job loss) | 8 (4.2) | 38 (7.3) | 0.119 | 0.53 | 0.24–1.18 | 0.212 | 0.46 | 0.13–1.56 |
| Have an income source (employed/Government benefits) | 96 (50.0) | 259 (50.0) | 0.680 | 0.93 | 0.66–1.31 | 0.893 | 0.92 | 0.29–2.98 |
| **Perceived distress due to change of employment status** | **189** | **510** | | | | | | |
| A little to none | 130 (68.8) | 352 (69.0) | | 1 | | | 1 | |
| Moderate to a great deal | 59 (31.2) | 158 (31.0) | 0.952 | 1.01 | 0.71–1.45 | NA | NA | NA |
| **Self-identification as a frontline or essential service worker** | **195** | **525** | | | | | | |
| No | 138 (70.8) | 366 (69.7) | | 1 | | | 1 | |
| Yes | 57 (29.2) | 159 (30.3) | 0.784 | 0.95 | 0.66–1.36 | 0.112 | 0.64 | 0.37–1.11 |
| **COVID-19 impacted financial situation** | **195** | **525** | | | | | | |
| No | 92 (47.2) | 287 (54.7) | | 1 | | | 1 | |
| Yes | 103 (52.8) | 238 (45.3) | 0.074 | 1.35 | 0.97–1.88 | 0.119 | 1.33 | 0.93–1.89 |
| **Co-morbidities** | **161** | **451** | | | | | | |
| No | 123 (76.4) | 355 (78.7) | | 1 | | | 1 | |
| Psychiatric/Mental health problem | 3 (1.9) | 17 (3.8) | 0.288 | 0.51 | 0.15–1.77 | 0.648 | 0.50 | 0.03–9.77 |

*(Continued)*

**Table 6.** (Continued)

| Characteristics | High (score 22–35), n(%) | Low (score 7–21), n(%) | Unadjusted analyses | | | Adjusted analyses | | |
|---|---|---|---|---|---|---|---|---|
| | | | p | OR | 95% CIs | p | AOR | 95% CIs |
| Other co-morbidities* | 35 (21.7) | 79 (17.5) | 0.282 | 1.28 | 0.82–2.00 | 0.585 | 0.81 | 0.37–1.75 |
| **Smoking** | **195** | **525** | | | | | | |
| Never smoker | 172 (88.2) | 484 (92.2) | | 1 | | | 1 | |
| Ever smoker (Daily/Non-daily/Ex) | 23 (11.8) | 41 (7.8) | 0.095 | 1.58 | 0.92–2.71 | 0.413 | 1.61 | 0.52–4.99 |
| **Current alcohol drinking (last 4 weeks)** | **193** | **520** | | | | | | |
| No | 186 (96.4) | 479 (92.1) | | 1 | | | 1 | |
| Yes | 7 (3.6) | 41 (7.9) | *0.044* | *0.44* | *0.19–0.99* | *0.006* | *0.26* | *0.10–0.68* |
| **Provided care to a family member/patient with known/suspected case of COVID-19** | **194** | **521** | | | | | | |
| No | 175 (90.2) | 472 (90.6) | | 1 | | | 1 | |
| Yes | 19 (9.8) | 49 (9.4) | 0.875 | 1.05 | 0.60–1.83 | 0.60 | 1.17 | 0.65–2.11 |
| **Experience related to COVID-19 pandemic** | **183** | **505** | | | | | | |
| No known exposure to COVID-19 | 164 (89.6) | 474 (93.9) | | 1 | | | 1 | |
| I had recent overseas travel history and was in self-quarantine | 4 (2.2) | 6 (1.2) | 0.314 | 1.93 | 0.54–6.91 | NA | NA | NA |
| I have been tested negative for COVID-19 but self-isolating | 15 (8.2) | 25 (5.0) | 0.104 | 1.73 | 0.89–3.37 | *0.042* | *3.12* | *1.04–9.32* |
| **Self-identification as a patient (visited a healthcare provider in the last 4 weeks)** | **194** | **521** | | | | | | |
| No | 132 (68.0) | 355 (68.1) | | 1 | | | 1 | |
| Yes | 62 (32.0) | 166 (31.9) | 0.980 | 1.00 | 0.71–1.43 | 0.924 | 1.02 | 0.70–1.48 |
| **Healthcare service use in the last 4 weeks** | **79** | **222** | | | | | | |
| Telehealth consultation/Use of national helpline | 12 (15.2) | 33 (14.9) | | 1 | | | 1 | |
| In-person visit to a healthcare provider | 61 (77.2) | 179 (80.6) | 0.860 | 0.94 | 0.46–1.93 | 0.413 | 0.73 | 0.34–1.57 |
| Used both services | 6 (7.6) | 10 (4.5) | 0.417 | 1.65 | 0.49–5.53 | 0.804 | 1.18 | 0.33–4.24 |
| **Level of psychological distress (K10 categories)** | **195** | **525** | | | | | | |
| Low (score 10–15) | 49 (25.1) | 224 (42.7) | | 1 | | | 1 | |
| Moderate to Very High (score 16–50) | 146 (74.9) | 301 (57.3) | *<0.001* | *2.22* | *1.54–3.20* | *<0.001* | *2.56* | *1.71–3.83* |
| **Level of coping (BRCS categories)** | **195** | **525** | | | | | | |
| Low resilient coping (score 4–13) | 77 (39.5) | 174 (33.1) | | 1 | | | 1 | |
| Medium to high resilient coping (score 14–20) | 118 (60.5) | 351 (66.9) | 0.112 | 0.76 | 0.54–1.07 | 0.074 | 0.72 | 0.50–1.03 |
| **Healthcare service use to overcome COVID-19 related stress in the last 4 weeks** | **191** | **516** | | | | | | |
| No | 185 (96.9) | 504 (97.7) | | 1 | | | 1 | |
| Yes | 6 (3.1) | 12 (2.3) | 0.541 | 1.36 | 0.50–3.68 | 0.453 | 1.47 | 0.54–4.02 |

Adjusted for: age, gender, living status, born in Malaysia, education and employment

* Cardiac diseases/Stroke/Hypertension/Hyperlipidemia/Diabetes/Cancer/Chronic respiratory disease

**Table 7. Factors associated with coping among the study population (based on BRCS scoring).**

| Characteristics | Medium to High (score 14–20), n(%) | Low (score 4–13), n(%) | Unadjusted analyses | | | Adjusted analyses | | |
|---|---|---|---|---|---|---|---|---|
| | | | p | OR | 95% CIs | p | AOR | 95% CIs |
| Total study participants | 469 | 251 | | | | | | |
| **Age groups** | **459** | **243** | | | | | | |
| 18–29 years | 250 (54.5) | 148 (60.9) | | 1 | | | 1 | |
| 30–59 years | 195 (42.5) | 87 (35.8) | 0.087 | 1.33 | 0.96–1.84 | 0.525 | 1.23 | 0.65–2.34 |
| ≥60 years | 14 (3.1) | 8 (3.3) | 0.938 | 1.04 | 0.42–2.53 | 0.831 | 0.88 | 0.27–2.84 |
| **Gender** | **468** | **250** | | | | | | |
| Male | 153 (32.7) | 82 (32.8) | | 1 | | | 1 | |
| Female | 315 (67.3) | 168 (67.2) | 0.977 | 1.01 | 0.73–1.39 | 0.455 | 0.76 | 0.38–1.55 |
| **Living status** | **454** | **241** | | | | | | |
| Live without family members (on your own/shared house/others) | 86 (18.9) | 50 (20.7) | | 1 | | | 1 | |
| Live with family members (partner and/or children) | 368 (81.1) | 191 (79.3) | 0.568 | 1.12 | 0.76–1.65 | 0.125 | 2.64 | 0.77–9.10 |
| **Born in Malaysia** | **469** | **251** | | | | | | |
| No | 48 (10.2) | 18 (7.2) | | 1 | | | 1 | |
| Yes | 421 (89.8) | 233 (92.8) | 0.175 | 0.68 | 0.39–1.19 | 0.032 | 0.33 | 0.12–0.91 |
| **Completed level of education** | **466** | **250** | | | | | | |
| Secondary | 77 (16.5) | 36 (14.4) | | 1 | | | 1 | |
| Diploma | 83 (17.8) | 36 (14.4) | 0.792 | 1.08 | 0.62–1.88 | 0.842 | 1.19 | 0.21–6.59 |
| Degree (Bachelor) | 192 (41.2) | 109 (43.6) | 0.408 | 0.82 | 0.52–1.30 | 0.899 | 0.88 | 0.13–5.90 |
| Masters and above | 114 (24.5) | 69 (27.6) | 0.308 | 0.77 | 0.47–1.27 | 0.083 | 0.39 | 0.13–1.13 |
| **Current employment condition** | **460** | **250** | | | | | | |
| Unemployed/Home duties | 201 (43.7) | 108 (43.2) | | 1 | | | 1 | |
| Jobs affected by COVID-19 (lost job/working hours reduced/afraid of job loss) | 29 (6.3) | 29 (6.3) | 0.791 | 0.92 | 0.48–1.74 | 0.103 | 0.32 | 0.08–1.26 |
| Have an income source (employed/Government benefits) | 230 (50.0) | 230 (50.0) | 0.944 | 0.99 | 0.72–1.36 | 0.320 | 0.49 | 0.12–1.99 |
| **Perceived distress due to change of employment status** | **450** | **249** | | | | | | |
| A little to none | 315 (70.0) | 167 (67.1) | | 1 | | | 1 | |
| Moderate to a great deal | 135 (30.0) | 82 (32.9) | 0.422 | 0.87 | 0.63–1.22 | NA | NA | NA |
| **Self-identification as a frontline or essential service worker** | **469** | **251** | | | | | | |
| No | 327 (69.7) | 177 (70.5) | | 1 | | | 1 | |
| Yes | 142 (30.3) | 74 (29.5) | 0.824 | 1.04 | 0.74–1.45 | 0.655 | 0.89 | 0.55–1.46 |
| **COVID-19 impacted financial situation** | **469** | **251** | | | | | | |
| No | 262 (55.9) | 117 (46.6) | | 1 | | | 1 | |
| Yes | 207 (44.1) | 134 (53.4) | *0.018* | *0.69* | *0.51–0.94* | 0.058 | 0.73 | 0.52–1.01 |
| **Co-morbidities** | **395** | **217** | | | | | | |
| No | 304 (77.0) | 174 (80.2) | | 1 | | | 1 | |
| Psychiatric/Mental health problem | 11 (2.8) | 9 (4.1) | 0.437 | 0.70 | 0.28–1.72 | 0.629 | 1.67 | 0.21–13.2 |

(*Continued*)

**Table 7.** (Continued)

| Characteristics | Medium to High (score 14–20), n(%) | Low (score 4–13), n(%) | Unadjusted analyses | | | Adjusted analyses | | |
|---|---|---|---|---|---|---|---|---|
| | | | p | OR | 95% CIs | p | AOR | 95% CIs |
| Other co-morbidities* | 80 (20.3) | 34 (15.7) | 0.187 | 1.35 | 0.87–2.10 | 0.391 | 1.34 | 0.21–13.2 |
| **Smoking** | **469** | **251** | | | | | | |
| Never smoker | 423 (90.2) | 233 (92.8) | | 1 | | | 1 | |
| Ever smoker (Daily/Non-daily/Ex) | 46 (9.8) | 18 (7.2) | 0.236 | 1.41 | 0.80–2.48 | 0.564 | 1.41 | 0.44–4.52 |
| **Current alcohol drinking (last 4 weeks)** | **466** | **247** | | | | | | |
| No | 434 (93.1) | 231 (93.5) | | 1 | | | 1 | |
| Yes | 32 (6.9) | 16 (6.5) | 0.844 | 1.07 | 0.57–1.98 | 0.813 | 1.08 | 0.56–2.09 |
| **Provided care to a family member/patient with known/ suspected case of COVID-19** | **465** | **250** | | | | | | |
| No | 415 (89.2) | 232 (92.8) | | 1 | | | 1 | |
| Yes | 50 (10.8) | 18 (7.2) | 0.123 | 1.55 | 0.89–2.73 | *0.046* | *1.87* | *1.01–3.46* |
| **Experience related to COVID-19 pandemic** | **442** | **246** | | | | | | |
| No known exposure to COVID-19 | 407 (92.1) | 231 (92.1) | | 1 | | | 1 | |
| I had recent overseas travel history and was in self-quarantine | 4 (0.9) | 6 (2.4) | 0.135 | 0.38 | 0.11–1.35 | NA | NA | NA |
| I have been tested negative for COVID-19 but self-isolating | 31 (7.0) | 9 (3.7) | 0.084 | 1.95 | 0.91–4.18 | 0.090 | 3.73 | 0.81–17.1 |
| **Self-identification as a patient (visited a healthcare provider in the last 4 weeks)** | **468** | **247** | | | | | | |
| No | 324 (69.2) | 163 (66.0) | | 1 | | | 1 | |
| Yes | 144 (30.8) | 84 (34.0) | 0.377 | 0.86 | 0.62–1.20 | 0.283 | 0.83 | 0.58–1.17 |
| **Healthcare service use in the last 4 weeks** | **184** | **177** | | | | | | |
| Telehealth consultation/Use of national helpline | 27 (14.7) | 18 (15.4) | | 1 | | | 1 | |
| In-person visit to a healthcare provider | 144 (78.3) | 96 (82.1) | 1.000 | 1.00 | 0.52–1.92 | 0.982 | 1.01 | 0.51–2.01 |
| Used both services | 13 (7.1) | 3 (2.6) | 0.135 | 2.89 | 0.72–11.6 | 0.133 | 3.02 | 0.71–12.8 |
| **Level of psychological distress (K10 categories)** | **469** | **251** | | | | | | |
| Low (score 10–15) | 198 (42.2) | 75 (29.9) | | 1 | | | 1 | |
| Moderate to Very High (score 16–50) | 271 (57.8) | 176 (70.1) | *0.001* | *0.58* | *0.42–0.81* | *0.001* | *0.54* | *0.38–0.76* |
| **Level of fear of COVID-19 (FCV-19S categories)** | **469** | **251** | | | | | | |
| Low (score 7–21) | 351 (74.8) | 174 (69.3) | | 1 | | | 1 | |
| High (score 22–35) | 118 (25.2) | 77 (30.7) | 0.112 | 0.76 | 0.54–1.07 | 0.074 | 0.72 | 0.50–1.03 |
| **Healthcare service use to overcome COVID-19 related stress in the last 4 weeks** | **465** | **242** | | | | | | |
| No | 456 (98.1) | 233 (96.3) | | 1 | | | 1 | |
| Yes | 9 (1.9) | 9 (3.7) | 0.153 | 0.51 | 0.20–1.31 | 0.193 | 0.53 | 0.21–1.37 |

Adjusted for: age, gender, living status, born in Malaysia, education and employment

* Cardiac diseases/Stroke/Hypertension/Hyperlipidemia/Diabetes/Cancer/Chronic respiratory disease

pandemic [41]. Our study has also indicated that low resilience was associated with moderate to high levels of psychological distress while moderate to high resilience was not only associated with lower psychological distress, but also enabled the individual to provide care to family members or patients infected with COVID-19. Hence, our study highlighted the pivotal role of resilience in overcoming the psychological impact of the COVID-19 pandemic.

The strength of this study was the use of validated tools to investigate the factors associated with psychological distress, fear and coping strategies in Malaysia. Due to nation-wide travel restrictions, online survey was the only feasible way for data collection, and we were able to recruit a large sample of Malaysian population during the critical pandemic period. However, there were some limitations in this study. As this study was an online survey, most younger people participated into this survey as they were more active on social media. The study was conducted in English, so those who were not well versed in English might not be able to take part in the study. It was beyond the scope of the study to check and ensure that the participants had sufficient ability in understanding English. Due to the self-reporting nature of the survey, possibility of reporting bias cannot be excluded. The survey responses were predominantly from west Malaysia, although the survey link was shared across all the states in Malaysia through various social media platforms and emails. This could be explained by the researchers' use of snowball sampling techniques which reflected their community acquaintances and accessibility to clinics/allied health service facilities more in West Malaysia than in the eastern part of Malaysia. Another important limitation of our study was, those who might have tested positive to COVID-19 or those whose family members or friends were tested positive with COVID-19 infection or who were interested to this topic were more likely to participate into this survey. We also acknowledge that we might have missed the more marginalized or vulnerable group of population in this study (e.g., those who were more isolated specially people from rural areas, from the areas of poor internet access, older people those who were not active in social media, or migrant or other minority groups); therefore, the findings of this study could be potentially underestimated and might not be representative to the general Malaysian Population.

## Conclusions

The study identified some of the key risk factors for developing psychological distress, fear and coping strategies during the COVID-19 pandemic in Malaysia. Vulnerable groups of individuals such as patients and those impacted financially during COVID-19 should be supported for their mental wellbeing. Behavioural interventions should be targeted to reduce the impact of alcohol drinking during such crisis period. Findings of this study would assist the researchers to plan future studies with vulnerable groups of Malaysians, specifically exploring the strategies to support their mental wellbeing during the pandemic and post-pandemic period. Specific interventions based on the emerging evidence arising from Malaysian and global studies can be tested to alleviate psychological distress, fear and improve resilience among Malaysian population.

## Acknowledgments

We would like to acknowledge the support from Trisha Zafrin, Nur Syakirarah Binti Mohamed Elias for helping us for data collection.

## Author Contributions

**Conceptualization:** Wendy Cross, Muhammad Aziz Rahman.

**Data curation:** Ahmed Suparno Bahar Moni, Shalimar Abdullah, Mohammad Farris Iman Leong Bin Abdullah, Mohammed Shahjahan Kabir, Muhammad Aziz Rahman.

**Investigation:** Ahmed Suparno Bahar Moni, Shalimar Abdullah, Mohammad Farris Iman Leong Bin Abdullah, Mohammed Shahjahan Kabir.

**Methodology:** Ahmed Suparno Bahar Moni, Muhammad Aziz Rahman.

**Project administration:** Ahmed Suparno Bahar Moni, Muhammad Aziz Rahman.

**Resources:** Ahmed Suparno Bahar Moni.

**Software:** Muhammad Aziz Rahman.

**Supervision:** Muhammad Aziz Rahman.

**Writing – original draft:** Ahmed Suparno Bahar Moni, Mohammad Farris Iman Leong Bin Abdullah, Muhammad Aziz Rahman.

**Writing – review & editing:** Ahmed Suparno Bahar Moni, Sheikh M. Alif, Farhana Sultana, Masudus Salehin, Sheikh Mohammed Shariful Islam, Wendy Cross, Muhammad Aziz Rahman.

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
