## [Decision Letter · Decision Letter 0]

19 Feb 2021

PONE-D-20-40506

Psychological distress, fear and coping among Malaysians during the COVID-19 pandemic

PLOS ONE

Dear Dr. Bahar Moni,

Thank you for submitting your manuscript to PLOS ONE. After careful consideration, we feel that it has merit but does not fully meet PLOS ONE’s publication criteria as it currently stands. Therefore, we invite you to submit a revised version of the manuscript that addresses the points raised during the review process.

We look forward to receiving your revised manuscript.

Kind regards,

Amir H. Pakpour, Ph.D.

Academic Editor

PLOS ONE

Journal Requirements:

2. Please clarify in your Methods section whether the questionnaire is published under a CC-BY license, or whether you obtained permission from the publisher to reproduce the questionnaire in this manuscript. Please explain any copyright or restrictions on this questionnaire" Ping Copyright Discussions team (copyrightdiscussions@plos.org) for follow up.

We note that one or more of the authors are employed by a commercial company: Telstra Health.

3.1. Please provide an amended Funding Statement declaring this commercial affiliation, as well as a statement regarding the Role of Funders in your study. If the funding organization did not play a role in the study design, data collection and analysis, decision to publish, or preparation of the manuscript and only provided financial support in the form of authors' salaries and/or research materials, please review your statements relating to the author contributions, and ensure you have specifically and accurately indicated the role(s) that these authors had in your study. You can update author roles in the Author Contributions section of the online submission form.

3.2. Please also provide an updated Competing Interests Statement declaring this commercial affiliation along with any other relevant declarations relating to employment, consultancy, patents, products in development, or marketed products, etc.  

4. Thank you for submitting the above manuscript to PLOS ONE. During our internal evaluation of the manuscript, we found significant text overlap between your submission and the following previously published works, some of which you are an author.

- https://globalizationandhealth.biomedcentral.com/articles/10.1186/s12992-020-00624-w

- https://www.researchsquare.com/article/rs-57952/v1

- https://www.mdpi.com/1660-4601/17/17/6206/htm

Please revise the manuscript to rephrase the duplicated text, cite your sources, and provide details as to how the current manuscript advances on previous work. Please note that further consideration is dependent on the submission of a manuscript that addresses these concerns about the overlap in text with published work.

Reviewers' comments:

Reviewer's Responses to Questions

**Comments to the Author**

1. Is the manuscript technically sound, and do the data support the conclusions?

Reviewer #1: Partly

2. Has the statistical analysis been performed appropriately and rigorously? 

Reviewer #1: Yes

3. Have the authors made all data underlying the findings in their manuscript fully available?

Reviewer #1: Yes

4. Is the manuscript presented in an intelligible fashion and written in standard English?

Reviewer #1: No

5. Review Comments to the Author

Reviewer #1: The study entitled “Psychological distress, fear and coping among Malaysians during the COVID-19 pandemic” investigated the psychological distress, fear and coping strategies as a result of the COVID-19 pandemic among Malaysian residents. Additionally, the factors associated with the psychological distress, fear, and coping strategies are studied and discussed. The sample size is good (n=720) for this study and the analyses seem appropriate (several logistic regression models). However, the presentation and the literature review of the present submission are unsatisfactory. The authors should improve their submission substantially to achieve the scientific rigor. Below please see my specific comments.

1. There are many papers on the impacts of COVID-19 on psychological distress. However, the authors did not review these papers properly in their Introduction. I suggest the authors take references from the following publications, which have outlined the psychological distress during COVID-19 in different populations, including children, older people, pregnant women, university students, people with overweight, online population, and general population. I believe that it is very important to enrich the Introduction.

Nathiya D, Singh P, Suman S, Raj P, Tomar BS. Mental health problems and impact on youth minds during the COVID-19 outbreak: Cross-sectional (RED-COVID) survey. Soc Health Behav 2020;3:83-8

Lin CY. Social reaction toward the 2019 novel coronavirus (COVID-19). Soc Health Behav 2020;3:1-2

Chen, C.-Y., Chen, I.-H., Pakpour, A. H., Lin, C.-Y., & Griffiths, M. D. (accepted). Internet-related behaviors and psychological distress among schoolchildren during the COVID-19 school hiatus. Cyberpsychology, Behavior, and Social Networking.

Chen, I.-H., Chen, C.-Y., Pakpour, A. H., Griffiths, M. D., Lin, C.-Y., Li, X.-D., Tsang, H. W. H. (2020). Problematic internet-related behaviors mediate the associations between levels of internet engagement and distress among schoolchildren during COVID-19 lockdown: A longitudinal structural equation modeling study. Journal of Behavioral Addictions. doi: 10.1556/2006.2021.00006

Pramukti, I., Strong, C., Sitthimongkol, Y., Setiawan, A., Pandin M. G. R., Yen, C.-F., Lin, C.-Y., Griffiths, M. D., Ko, N.-Y. (2020). Anxiety and suicidal thoughts during the COVID-19 pandemic: A cross-country comparison among Indonesian, Taiwanese, and Thai university students. Journal of Medical Internet Research, 22(12), e24487.

Chen, C.-Y., Chen, I.-H., O'Brien, K. S., Latner, J. D., & Lin, C.-Y. (2020). Psychological distress and internet-related behaviors between schoolchildren with and without overweight during the COVID-19 outbreak. International Journal of Obesity. https://doi.org/10.1038/s41366-021-00741-5

Fazeli, S., Zeidi, I. M., Lin, C.-Y., Namdar, P., Griffiths, M. D., Ahorsu, D. K., Pakpour, A. H. (2020). Depression, anxiety, and stress mediate the associations between internet gaming disorder, insomnia, and quality of life during the COVID-19 outbreak. Addictive Behaviors Reports, 12, 100307.

Mamun, M. A., Sakib, N., Gozal, D., Bhuiyan, A. I., Hossain, S., Bodrud-Doza, M., Mamun, F. A., Hosen, I., Abdullah, A. H., Sarker, M. A., Rayhan, I., Sikder, M. T., Muhit, M., Lin, C.-Y., Griffiths, M. D., Pakpour, A. H. (2021). The COVID-19 pandemic and serious psychological consequences in Bangladesh: a population-based nationwide study. Journal of Affective Disorders, 279, 462-472.

Hashemi, S. G. S., Hosseinnezhad, S., Dini, S., Griffiths, M. D., Lin, C.-Y., Pakpour, A. H. (2020). The mediating effect of the cyberchondria and anxiety sensitivity in the association between problematic internet use, metacognition belief and fear of COVID-19 among Iranian online population. Heliyon, 6(10), e05135.

Ahorsu, D. K., Lin, C.-Y., Pakpour, A. H. (2020). The association between health status and insomnia, mental health, and preventive behaviours: The mediating role of fear of COVID-19. Gerontology and Geriatric Medicine, 6, 1-9.

Lin, C.-Y., Broström, A., Griffiths, M. D., & Pakpour, A. H. (2020). Investigating mediated effects of fear of COVID-19 and COVID-19 misunderstanding in the association between problematic social media use and distress/insomnia. Internet Interventions, 21, 100345.

Chen, I.-H., Chen, C.-Y., Pakpour, A. H., Griffiths, M. D., & Lin, C.-Y. (2020). Internet-related behaviors and psychological distress among schoolchildren during COVID-19 school suspension. Journal of the American Academy of Child and Adolescent Psychiatry, 159(10), 1099-1102.

Ahorsu, D. K., Imani, V., Lin, C.-Y., Timpka, T., Broström, A., Updegraff, J. A., Årestedt, K., Griffiths, M. D., Pakpour, A. H. (accepted). Associations between fear of COVID-19, mental health, and preventive behaviours across pregnant women and husbands: An actor-partner interdependence modelling. International Journal of Mental Health and Addiction.

Chen, C.-Y., Chen, I.-H., Pakpour, A. H., Lin, C.-Y., & Griffiths, M. D. (accepted). Internet-related behaviors and psychological distress among schoolchildren during the COVID-19 school hiatus. Cyberpsychology, Behavior, and Social Networking

2. Given the references I have provided in the comment #1, I think that the statement “However, there has been limited evidence regarding the impact of COVID-19 on community members including healthcare workers” is not supported. Please note that I only provide some references, and there are ample references in the literature that I have not listed.

3. The Methods part provides little information regarding how the authors distribute the online survey. How did the authors use these online platforms to recruit potential participants?

4. How could the authors identify whether the participants were capable of responding to an online questionnaire in English? Specifically, any participant whose English is poor still can answer the English online survey. What the participant has to do is only to click on the answers in random. I wonder how the authors control this factor. Using answering time within 1 minute may work, but not really reflect because the participant whose English is poor still can use a lot of time in answering the online survey.

5. The authors said that they used “Snowball sampling technique”, please describe clearly how this technique was used in the online survey.

6. After I read the Data Collection subsection, I found that some of my queries above are answered in the subsection. However, this indicates that the authors did not arrange the presentation in a good manner. Therefore, the authors should think about how to clearly deliver the information in their revised manuscript.

7. The Study tool subsection is poorly written. The authors should still provide the information on the used instruments. It is quite irresponsive to refer the readers to a previous paper. Specifically, not all readers have time or access to the previous paper. I also wonder why the online survey made no changes given that cultures between Malaysia and Australia are different.

8. Following the previous comment, the authors did not provide proper citations to acknowledge the use of the studied instruments. For example, the citation of Ahorsu et al. (accepted) should be acknowledged for the use of FCV-19S. Similar issues are happened to other used instruments.

Reference:

Ahorsu, D. K., Lin, C.-Y., Imani, V., Saffari, M., Griffiths, M. D., & Pakpour, A. H. (accepted). Fear of COVID-19 Scale: Development and initial validation. International Journal of Mental Health and Addiction

9. The authors should provide proper citations to indicate their used cutoffs for every studied instruments.

10. When reporting the significance, please do not use p=0.000. Use p<0.001 instead given that the p-value will never be 0.

6. PLOS authors have the option to publish the peer review history of their article (what does this mean?). If published, this will include your full peer review and any attached files.

Reviewer #1: No

---

## [Author Response · Author response to Decision Letter 0]

30 Mar 2021

Response to reviewers

Psychological distress, fear and coping among Malaysians during the COVID-19 pandemic (PONE-D-20-40506)

The line number and page numbers are mentioned according to the track change version of the manuscript

Comments from the Editor with Responses

Response: The manuscript has been re-formatted accordingly.

2. Please clarify in your Methods section whether the questionnaire is published under a CC-BY license, or whether you obtained permission from the publisher to reproduce the questionnaire in this manuscript.

Response: The questionnaire was not published in the earlier study. Therefore, obtaining permission from the publisher was not applicable. The lead investigator of this study (Rahman MA) was also the lead investigator for the Australian study along with a number of investigators common in both studies. Therefore, no further permission was deemed necessary to use the same study tool for this study. It has already been mentioned: (page-11, line-09)

“We used the same survey questionnaire (except residence location/region in Malaysia) which was used earlier by the Australian investigators included in this study [7].”

We note that one or more of the authors are employed by a commercial company: Telstra Health.

Response: This has been addressed and included in the cover letter as advised:

In terms of funding, I can certify on behalf of all the authors that we did not receive funding for this study. Dr Farhana Sultana is a staff of Telstra Health, but the commercial organization did not fund this study and did not have any role in this study. 

In addition, I can confirm that we did not have any competing interests. The commercial affiliation does not alter our adherence to PLOS ONE policies on sharing data and materials.

In addition, we have also updated the manuscript accordingly (Page-19).

4. Thank you for submitting the above manuscript to PLOS ONE. During our internal evaluation of the manuscript, we found significant text overlap between your submission and the following previously published works, some of which you are an author. Please revise the manuscript to rephrase the duplicated text, cite your sources, and provide details as to how the current manuscript advances on previous work.

Response: We have rephrased the entire manuscript, revised and added citations as needed.

In terms of what this study adds further to the existing evidence, we have explained this in the revised Introduction section. 

Comments from the Reviewer-1 

The study entitled “Psychological distress, fear and coping among Malaysians during the COVID-19 pandemic” investigated the psychological distress, fear and coping strategies as a result of the COVID-19 pandemic among Malaysian residents. Additionally, the factors associated with the psychological distress, fear, and coping strategies are studied and discussed. The sample size is good (n=720) for this study and the analyses seem appropriate (several logistic regression models). However, the presentation and the literature review of the present submission are unsatisfactory. The authors should improve their submission substantially to achieve the scientific rigor. Below please see my specific comments.

Response: We would like to thank the reviewer for the kind feedback.

1. There are many papers on the impacts of COVID-19 on psychological distress. However, the authors did not review these papers properly in their Introduction. I suggest the authors take references from the following publications, which have outlined the psychological distress during COVID-19 in different populations, including children, older people, pregnant women, university students, people with overweight, online population, and general population. I believe that it is very important to enrich the Introduction:

o Nathiya D, Singh P, Suman S, Raj P, Tomar BS. Mental health problems and impact on youth minds during the COVID-19 outbreak: Cross-sectional (RED-COVID) survey. Soc Health Behav 2020;3:83-8

o Lin CY. Social reaction toward the 2019 novel coronavirus (COVID-19). Soc Health Behav 2020;3:1-2

o Chen, C.-Y., Chen, I.-H., Pakpour, A. H., Lin, C.-Y., & Griffiths, M. D. (accepted). Internet-related behaviors and psychological distress among schoolchildren during the COVID-19 school hiatus. Cyberpsychology, Behavior, and Social Networking.

o Chen, I.-H., Chen, C.-Y., Pakpour, A. H., Griffiths, M. D., Lin, C.-Y., Li, X.-D., Tsang, H. W. H. (2020). Problematic internet-related behaviors mediate the associations between levels of internet engagement and distress among schoolchildren during COVID-19 lockdown: A longitudinal structural equation modeling study. Journal of Behavioral Addictions. doi: 10.1556/2006.2021.00006

o Pramukti, I., Strong, C., Sitthimongkol, Y., Setiawan, A., Pandin M. G. R., Yen, C.-F., Lin, C.-Y., Griffiths, M. D., Ko, N.-Y. (2020). Anxiety and suicidal thoughts during the COVID-19 pandemic: A cross-country comparison among Indonesian, Taiwanese, and Thai university students. Journal of Medical Internet Research, 22(12), e24487.

o Chen, C.-Y., Chen, I.-H., O'Brien, K. S., Latner, J. D., & Lin, C.-Y. (2020). Psychological distress and internet-related behaviors between schoolchildren with and without overweight during the COVID-19 outbreak. International Journal of Obesity. https://doi.org/10.1038/s41366-021-00741-5

o Fazeli, S., Zeidi, I. M., Lin, C.-Y., Namdar, P., Griffiths, M. D., Ahorsu, D. K., Pakpour, A. H. (2020). Depression, anxiety, and stress mediate the associations between internet gaming disorder, insomnia, and quality of life during the COVID-19 outbreak. Addictive Behaviors Reports, 12, 100307.

o Mamun, M. A., Sakib, N., Gozal, D., Bhuiyan, A. I., Hossain, S., Bodrud-Doza, M., Mamun, F. A., Hosen, I., Abdullah, A. H., Sarker, M. A., Rayhan, I., Sikder, M. T., Muhit, M., Lin, C.-Y., Griffiths, M. D., Pakpour, A. H. (2021). The COVID-19 pandemic and serious psychological consequences in Bangladesh: a population-based nationwide study. Journal of Affective Disorders, 279, 462-472.

o Hashemi, S. G. S., Hosseinnezhad, S., Dini, S., Griffiths, M. D., Lin, C.-Y., Pakpour, A. H. (2020). The mediating effect of the cyberchondria and anxiety sensitivity in the association between problematic internet use, metacognition belief and fear of COVID-19 among Iranian online population. Heliyon, 6(10), e05135.

o Ahorsu, D. K., Lin, C.-Y., Pakpour, A. H. (2020). The association between health status and insomnia, mental health, and preventive behaviours: The mediating role of fear of COVID-19. Gerontology and Geriatric Medicine, 6, 1-9.

o Lin, C.-Y., Broström, A., Griffiths, M. D., & Pakpour, A. H. (2020). Investigating mediated effects of fear of COVID-19 and COVID-19 misunderstanding in the association between problematic social media use and distress/insomnia. Internet Interventions, 21, 100345.

o Chen, I.-H., Chen, C.-Y., Pakpour, A. H., Griffiths, M. D., & Lin, C.-Y. (2020). Internet-related behaviors and psychological distress among schoolchildren during COVID-19 school suspension. Journal of the American Academy of Child and Adolescent Psychiatry, 159(10), 1099-1102.

o Ahorsu, D. K., Imani, V., Lin, C.-Y., Timpka, T., Broström, A., Updegraff, J. A., Årestedt, K., Griffiths, M. D., Pakpour, A. H. (accepted). Associations between fear of COVID-19, mental health, and preventive behaviours across pregnant women and husbands: An actor-partner interdependence modelling. International Journal of Mental Health and Addiction.

o Chen, C.-Y., Chen, I.-H., Pakpour, A. H., Lin, C.-Y., & Griffiths, M. D. (accepted). Internet-related behaviors and psychological distress among schoolchildren during the COVID-19 school hiatus. Cyberpsychology, Behavior, and Social Networking

Response: We would like to thank the reviewer for this suggestion. In fact, there had been a surge of publications on COVID-19 since we drafted and submitted it to the journal. Based on the feedback from the reviewer and suggested citations, we have carefully reviewed and revised the entire Introduction section by incorporating the relevant references.

2. Given the references I have provided in the comment #1, I think that the statement “However, there has been limited evidence regarding the impact of COVID-19 on community members including healthcare workers” is not supported. Please note that I only provide some references, and there are ample references in the literature that I have not listed.

Response: We do acknowledge the feedback and have updated the sentence with supporting evidence in the Introduction section (Page-9, line-5):

“There is limited evidence regarding the impact of COVID-19 on psychological distress, fear and coping strategies as a whole and amongst community members and healthcare workers in Malaysia.”

3. The Methods part provides little information regarding how the authors distribute the online survey. How did the authors use these online platforms to recruit potential participants?

Response: It has been addressed: (page-9, line-21)

“An online survey link was shared in different online platforms, including Facebook, Twitter and LinkedIn inviting online users to participate in this study.”

4. How could the authors identify whether the participants were capable of responding to an online questionnaire in English? Specifically, any participant whose English is poor still can answer the English online survey. What the participant has to do is only to click on the answers in random. I wonder how the authors control this factor. Using answering time within 1 minute may work, but not really reflect because the participant whose English is poor still can use a lot of time in answering the online survey.

Response: There was no practical way to control the issue that the reviewer has indicated. We have modified the terms used: (page-10, line-4)

“To be eligible, participants had to be 18 years or above and were literate enough to respond to an online questionnaire in English.”

5. The authors said that they used “Snowball sampling technique”, please describe clearly how this technique was used in the online survey.

Response: We have clarified this further: (pgae-10, line-13)

“Once any participant filled up the online questionnaire, he/she forwarded the survey link to own personal/professional networks.”

6. After I read the Data Collection subsection, I found that some of my queries above are answered in the subsection. However, this indicates that the authors did not arrange the presentation in a good manner. Therefore, the authors should think about how to clearly deliver the information in their revised manuscript.

Response: We have reviewed and updated the manuscript accordingly.

7. The Study tool subsection is poorly written. The authors should still provide the information on the used instruments. It is quite irresponsive to refer the readers to a previous paper. Specifically, not all readers have time or access to the previous paper. I also wonder why the online survey made no changes given that cultures between Malaysia and Australia are different.

Response: We have updated the section: (page-11, line-9)

“Three validated tools were included in the survey questionnaire. The Kessler Psychological Distress Scale (K10) tool having ten items was used to assess psychological distress [25], the Fear of COVID-19 scale (FCV-19S) having seven items was used to assess the levels of fear [26], and the Brief Resilient Coping Scale (BRCS) having four items was used to assess the levels of coping [27]. Each of those tools collected responses using a 5-point likert scale and the scoring was categorised as discussed in earlier study [7]. Reliability of using these tools had also been examined in a recent study [28].”

This study did not have any aim to assess cultural difference or impact of culture on the issues of our interest examined in this study. Therefore, the same questionnaire was used for Australia and Malaysia.

8. Following the previous comment, the authors did not provide proper citations to acknowledge the use of the studied instruments. For example, the citation of Ahorsu et al. (accepted) should be acknowledged for the use of FCV-19S. Similar issues are happened to other used instruments.

Reference:

Ahorsu, D. K., Lin, C.-Y., Imani, V., Saffari, M., Griffiths, M. D., & Pakpour, A. H. (accepted). Fear of COVID-19 Scale: Development and initial validation. International Journal of Mental Health and Addiction

Response: We have addressed this and updated the sub-section of study tool as described above.

9. The authors should provide proper citations to indicate their used cut offs for every studied instruments.

Response: We used the cut-offs as used in previously published studies, which has been mentioned in the sub-section of study tool: (page-11, line-13)

“Each of those tools collected responses using a 5-point likert scale and the scoring was categorised as discussed in earlier study [7].”

10. When reporting the significance, please do not use p=0.000. Use p<0.001 instead given that the p-value will never be 0.

Response: We have updated this accordingly.

---

## [Decision Letter · Decision Letter 1]

6 Apr 2021

PONE-D-20-40506R1

Psychological distress, fear and coping among Malaysians during the COVID-19 pandemic

PLOS ONE

Dear Dr. Bahar Moni,

Thank you for submitting your manuscript to PLOS ONE. After careful consideration, we feel that it has merit but does not fully meet PLOS ONE’s publication criteria as it currently stands. Therefore, we invite you to submit a revised version of the manuscript that addresses the points raised during the review process.

We look forward to receiving your revised manuscript.

Kind regards,

Amir H. Pakpour, Ph.D.

Academic Editor

PLOS ONE

Journal Requirements:

Reviewers' comments:

Reviewer's Responses to Questions

**Comments to the Author**

1. If the authors have adequately addressed your comments raised in a previous round of review and you feel that this manuscript is now acceptable for publication, you may indicate that here to bypass the “Comments to the Author” section, enter your conflict of interest statement in the “Confidential to Editor” section, and submit your "Accept" recommendation.

Reviewer #1: All comments have been addressed

2. Is the manuscript technically sound, and do the data support the conclusions?

Reviewer #1: Yes

3. Has the statistical analysis been performed appropriately and rigorously? 

Reviewer #1: Yes

4. Have the authors made all data underlying the findings in their manuscript fully available?

Reviewer #1: Yes

5. Is the manuscript presented in an intelligible fashion and written in standard English?

Reviewer #1: Yes

6. Review Comments to the Author

Reviewer #1: I would like to thank the reviewers that they have sincerely take my prior comments into consideration to prepare this revision. I can observe that the revised manuscript is substantially improved. I have only one minor comment for the authors to further address. That is, the authors acknowledged that they were unable to make sure that all the participants having sufficient ability in understanding English. Then, this should be listed as one of the limitations.

7. PLOS authors have the option to publish the peer review history of their article (what does this mean?). If published, this will include your full peer review and any attached files.

Reviewer #1: No

---

## [Author Response · Author response to Decision Letter 1]

11 Apr 2021

Dear Sir,

We would like to thank you for the kind feedback.

Your comments have been addressed and we have clarified this in the limitation section of the manuscript (page-16, line-17):

“The study was conducted in English, so those who were not well versed in English might not be able to take part in the study. It was beyond the scope of the study to check and ensure that the participants had sufficient ability in understanding English.”

---

## [Decision Letter · Decision Letter 2]

2 Aug 2021

PONE-D-20-40506R2

Psychological distress, fear and coping among Malaysians during the COVID-19 pandemic

PLOS ONE

Dear Authors,

Thank you for submitting your manuscript to PLOS ONE. After careful consideration, we feel that it has merit but does not fully meet PLOS ONE’s publication criteria as it currently stands. Therefore, we invite you to submit a revised version of the manuscript that addresses the points raised during the review process.

We look forward to receiving your revised manuscript.

Kind regards,

Alessio Gori, Ph. D.

Academic Editor

PLOS ONE

Journal Requirements:

Reviewers' comments:

Reviewer's Responses to Questions

**Comments to the Author**

1. If the authors have adequately addressed your comments raised in a previous round of review and you feel that this manuscript is now acceptable for publication, you may indicate that here to bypass the “Comments to the Author” section, enter your conflict of interest statement in the “Confidential to Editor” section, and submit your "Accept" recommendation.

Reviewer #1: All comments have been addressed

Reviewer #2: (No Response)

2. Is the manuscript technically sound, and do the data support the conclusions?

Reviewer #1: Yes

Reviewer #2: Yes

3. Has the statistical analysis been performed appropriately and rigorously? 

Reviewer #1: Yes

Reviewer #2: Yes

4. Have the authors made all data underlying the findings in their manuscript fully available?

Reviewer #1: Yes

Reviewer #2: No

5. Is the manuscript presented in an intelligible fashion and written in standard English?

Reviewer #1: Yes

Reviewer #2: Yes

6. Review Comments to the Author

Reviewer #1: The authors have now addressed my final concern and I am pleased to see this valuable paper to be published.

Reviewer #2: Manuscript ID: PONE-D-20-40506R2

1. Recommendation

Minor revision

2. Comments to Author:

Thank you for the opportunity of review this study entitled “Psychological distress, fear and coping among Malaysians during the COVID-19 pandemic”. The manuscript presented an investigation about the levels of psychological distress, fear and coping strategies and other related factors among Malaysian residents. The framework of this work is the COVID-19 pandemic.

In my opinion, the study has some relevant and interesting results. I think it's a good paper, well done. I commend the authors and other reviewers who have worked to reach this optimum point. There are only a few very minor issues that need to be addressed before the document is suitable for publication.

• Authors should be more detailed in the limitation section. The use of the online survey implies the need to pay attention in generalize results to general population, since people who do not have internet access (pronominally older individuals, of course) could be unrepresented. Furthermore, participants were voluntary, so only those who have interest in this research topic decided to participate: they may be unrepresentative of general Malaysian population.

• There are no incentives for future research: it is important not only to find a link with past results, but also to offer possible ideas for future work, in order to favour a continuous development of research. Therefore, please add implications for future research

• Parallelly, the paper could benefit from a focus on its strengths, highlighting the relevance of this research topic and results.

• In the conclusion section, the authors should focus on the practical implications of this study. Some studies provided an understanding of the treat responses at the time of COVID-19, as well as the effect of stress on health. Given these outcomes and the large incidence of these variables highlighted in Malaysian residents, the importance of effective tailor-made interventions on protective/risk variables (e.g., coping, defences etc.. based on previous studies) could be highlighted. I think the authors have a wide choice, given the large amount of literature available on this field. Therefore, the results of this study highlight the importance of working on some outcames (e.g., distress and fear) related to COVID-19 in Malaysian residents, preparing interventions that work on other associated variables, both those included in the study and other explored in previous research.

7. PLOS authors have the option to publish the peer review history of their article (what does this mean?). If published, this will include your full peer review and any attached files.

Reviewer #1: No

Reviewer #2: No

---

## [Author Response · Author response to Decision Letter 2]

6 Aug 2021

Response to reviewers

Psychological distress, fear and coping among Malaysians during the COVID-19 pandemic (PONE-D-20-40506R2)

The line number and page numbers are mentioned according to the track change version of the manuscript

Comments from the Reviewer-2 Responses

Thank you for the opportunity of review this study entitled “Psychological distress, fear and coping among Malaysians during the COVID-19 pandemic”. The manuscript presented an investigation about the levels of psychological distress, fear and coping strategies and other related factors among Malaysian residents. The framework of this work is the COVID-19 pandemic.

In my opinion, the study has some relevant and interesting results. I think it's a good paper, well done. I commend the authors and other reviewers who have worked to reach this optimum point. There are only a few very minor issues that need to be addressed before the document is suitable for publication.

Ans: We would like to thank the reviewer for the kind feedback.

The minor comments have been addressed in the revised version.

Authors should be more detailed in the limitation section. The use of the online survey implies the need to pay attention in generalize results to general population, since people who do not have internet access (pronominally older individuals, of course) could be unrepresented. Furthermore, participants were voluntary, so only those who have interest in this research topic decided to participate: they may be unrepresentative of general Malaysian population.

Ans: We have addressed this and expanded the limitation section further: 

(Page:16, Line: 22)

Due to the self-reporting nature of the survey, possibility of reporting bias cannot be excluded.

(Page: 17, Line: 04)

Another important limitation of our study was, those who might have tested positive to COVID-19 or those whose family members or friends were tested positive with COVID-19 infection or who were interested to this topic were more likely to participate into this survey. We also acknowledge that we might have missed the more marginalized or vulnerable group of population in this study (e.g., those who were more isolated specially people from rural areas, from the areas of poor internet access, older people those who were not active in social media, or migrant or other minority groups); therefore, the findings of this study could be potentially underestimated and might not be representative to the general Malaysian Population.

There are no incentives for future research: it is important not only to find a link with past results, but also to offer possible ideas for future work, in order to favour a continuous development of research. Therefore, please add implications for future research

Ans: We have added future research areas (Page: 17, Line: 21)

Findings of this study would assist the researchers to plan future studies with vulnerable groups of Malaysians, specifically exploring the strategies to support their mental wellbeing during the pandemic and post-pandemic period.

Parallelly, the paper could benefit from a focus on its strengths, highlighting the relevance of this research topic and results.

Ans: We have added strengths (Page:16, Line: 13)

The strength of this study was the use of validated tools to investigate the factors associated with psychological distress, fear and coping strategies in Malaysia. Due to nation-wide travel restrictions, online survey was the only feasible way for data collection, and we were able to recruit a large sample of Malaysian population during the critical pandemic period.

In the conclusion section, the authors should focus on the practical implications of this study. Some studies provided an understanding of the treat responses at the time of COVID-19, as well as the effect of stress on health. Given these outcomes and the large incidence of these variables highlighted in Malaysian residents, the importance of effective tailor-made interventions on protective/risk variables (e.g., coping, defences etc.. based on previous studies) could be highlighted. I think the authors have a wide choice, given the large amount of literature available on this field. Therefore, the results of this study highlight the importance of working on some outcames (e.g., distress and fear) related to COVID-19 in Malaysian residents, preparing interventions that work on other associated variables, both those included in the study and other explored in previous research

Ans: We have already included implications of current findings (Page:17, Line:18)

Vulnerable groups of individuals such as patients and those impacted financially during COVID-19 should be supported for their mental wellbeing. Behavioural interventions should be targeted to reduce the impact of alcohol drinking during such crisis period.

We have also added another implication (Page: 17, Line: 24)

Specific interventions based on the emerging evidence arising from Malaysian and global studies can be tested to alleviate psychological distress, fear and improve resilience among Malaysian population.

---

## [Editor Report · Decision Letter 3]

31 Aug 2021

Psychological distress, fear and coping among Malaysians during the COVID-19 pandemic

PONE-D-20-40506R3

Dear Dr. Ahmed Suparno Bahar Moni, 

We’re pleased to inform you that your manuscript has been judged scientifically suitable for publication and will be formally accepted for publication once it meets all outstanding technical requirements.

Kind regards,

Alessio Gori, Ph. D.

Academic Editor

PLOS ONE

---

## [Editor Report · Acceptance letter]

3 Sep 2021

PONE-D-20-40506R3 

Psychological distress, fear and coping among Malaysians during the COVID-19 pandemic 

Dear Dr. Bahar Moni:

I'm pleased to inform you that your manuscript has been deemed suitable for publication in PLOS ONE. Congratulations! Your manuscript is now with our production department. 

Kind regards, 

on behalf of

Dr. Alessio Gori 

Academic Editor

PLOS ONE